# Amyloid-β Effects on Peripheral Nerve: A New Model System

**DOI:** 10.3390/ijms241914488

**Published:** 2023-09-23

**Authors:** Mark M. Stecker, Ankita Srivastava, Allison B. Reiss

**Affiliations:** 1Fresno Institute of Neuroscience, Fresno, CA 93730, USA; 2Department of Medicine and Biomedical Research Institute, NYU Grossman Long Island School of Medicine, Long Island, NY 11501, USA; ankita.srivastava@nyulangone.org (A.S.); allison.reiss@nyulangone.org (A.B.R.)

**Keywords:** amyloid β-peptide, neuron, axon, sciatic nerve, action potential, peripheral nerve stimulation

## Abstract

Although there are many biochemical methods to measure amyloid-β (Aβ)42 concentration, one of the critical issues in the study of the effects of Aβ42 on the nervous system is a simple physiological measurement. The in vitro rat sciatic nerve model is employed and the nerve action potential (NAP) is quantified with different stimuli while exposed to different concentrations of Aβ42. Aβ42 predominantly reduces the NAP amplitude with minimal effects on other parameters except at low stimulus currents and short inter-stimulus intervals. The effects of Aβ42 are significantly concentration-dependent, with a maximum reduction in NAP amplitude at a concentration of 70 nM and smaller effects on the NAP amplitude at higher and lower concentrations. However, even physiologic concentrations in the range of 70 pM did reduce the NAP amplitude. The effects of Aβ42 became maximal 5–8 h after exposure and did not reverse during a 30 min washout period. The in vitro rat sciatic nerve model is sensitive to the effects of physiologic concentrations of Aβ42. These experiments suggest that the effect of Aβ42 is a very complex function of concentration that may be the result of amyloid-related changes in membrane properties or sodium channels.

## 1. Introduction

The accumulation of the amyloid fragment Aβ42 [1,2,3] is an important part of the pathophysiology of Alzheimer’s disease, although it is likely not the only process contributing to clinical manifestations [4]. Thus, it is important to be able to quantify its effects. The concentration of Aβ42 can be quantified biochemically [5,6], but this requires a sampling of body fluids. In vivo imaging quantitates total amyloid [7,8,9,10] but has difficulty distinguishing between different fragments [11]. One alternative is to quantify the physiologic effects of Aβ42 in a model system. Measuring the physiologic effects of Aβ42 is difficult because these effects are dependent not only on its concentration and the concentrations of possible ligands but on its conformation and state of aggregation [12,13]. Most commonly, the physiologic effects of Aβ42 are extrapolated from studies of cognitive function in humans [14] or animal models [15]. However, such testing is influenced by many environmental factors and can demonstrate significant variability [16]. Thus, a more objective, neurophysiological technique may help elucidate some of the effects of Aβ42. Neurophysiologic studies have investigated the role of Aβ42 at the single cell level using current clamp studies of individual neurons [17]. Other studies have investigated the effects of Aβ42 on hippocampal excitability [18,19], synaptic transmission [20,21,22], synaptic plasticity [23], and hippocampal gamma oscillations [24], among many more. These are complex model systems, and a simpler model system may possess many advantages as an assay.

The in vitro sciatic nerve model is a simple and well-studied system which has been useful in the study of the effects of many interventions, ranging from neurotoxicity [25,26] to studies on anoxia [27,28,29,30] and hyperglycemia [31,32]. Especially since the peripheral nerve is known to be affected by the systemic amyloidoses [33], it is conceivable that the in vitro sciatic nerve model may be sensitive to the effects of Aβ42. It is also possible that this model is sensitive to agents that modulate the production of Aβ42. In particular, β-secretase is involved in the control of peripheral nerve myelination [34] and can affect both voltage-gated potassium channels [35] and voltage-gated sodium channels [36,37,38]. In addition, murine studies have shown a link between Alzheimer’s disease and peripheral neuropathy in which the overexpression of APP results in dysfunction of both small and large peripheral nerve fibers [39]. Further, Aβ is associated with hearing loss, likely due to effects on the auditory nerve [40]. A reduced acoustic startle response and peripheral hearing loss are present in the 5xFAD mouse model of Alzheimer’s disease [41].

The specific purpose of this paper is to explore the possibility that the in vitro sciatic nerve model can be used to study the effects of Aβ42 on the peripheral nerve. This model system has the advantage that it has been studied extensively and the nerve action potential (NAP) provides a sensitive, easy to obtain, and easily quantifiable marker of the function of the nerve. This may allow testing of the physiologic effects of various interventions quickly in functioning axons and provide mechanistic information that complements biochemical, genomic, and proteomic data.

## 2. Results

### 2.1. Effects of Aβ42 on the NAP

#### 2.1.1. Parametric Data

Figure 1 shows the non-linear effects of Aβ42 concentration on the peak-to-peak amplitude of the first NAP in the second set for EXPTTIME = 36 (the end of the experiment, ISI = 166 ms, 15 mA stimulus current). At low concentrations, the amplitude of the NAP declines with increasing Aβ42 concentration up to 70 nM. Increasing the concentration of Aβ42 in the range 70 nM-70 μM increases the amplitude of the NAP. Above 70 uM, the amplitude of the NAP remains similar to its value without Aβ42. The ANOVA (Appendix A) demonstrates that this effect is statistically significant (F (8103) = 3.08, *p* = 0.004). This is confirmed by the Kruskal–Wallis test (H (8112) = 21.2, *p* = 0.007). The regression analysis demonstrates significant effects of CONC (*p* = 0.0006) and CONC*CONC (*p* = 0.001) (Appendix A). None of the other parameters derived from the NAP showed a statistically significant relationship with the Aβ42 concentration except for the peak amplitude (Appendix A).

The changes in the NAP over time and with the different stimuli do provide useful information. Figure 2 and Figure 3 show the time course of changes in the NAP amplitude as a function of ISI and stimulus current, respectively. In all conditions, the amplitude declines over time as the result of Wallerian degeneration. The effect of ISI is relatively consistent from 1.5 ms to 8 ms in the baseline and for each concentration of amyloid. The 1 ms ISI yields a very distorted waveform even in the absence of amyloid and the NAP amplitude declines more rapidly in all conditions over time than for the longer ISI values. For Aβ42 concentrations >700 pM and <700 µM, the effect of Aβ42 is greater at the lowest stimulus currents than the highest stimulus currents. This suggests that the effect of Aβ42 may be to reduce excitability in that concentration range. 

The analysis of parameters that are significantly correlated with Aβ42 concentration in the range of concentrations below 70 nM confirms that it is primarily the peak-to-peak amplitude of the NAP that is affected by Aβ42, as well as the peak amplitude and to a lesser extent the trough amplitude (Figure 4, Appendix A). Similar effects are seen if the criteria for choosing *p* values is *p* < 01 rather than FDR < 0.05. The χ^2^ analysis demonstrates that the presence of significant *p*-values mainly for the amplitude measurements is statistically significant (χ^2^ = 98, df = 66, *p* < 0.006 for the first stimulus and χ^2^ = 226, df = 66, *p* < 0.0001 for the second set of stimuli). The number of significant parameter relationships increased during the experiment and reached its maximum by roughly the 20th time period (Appendix A). It is important to note that contrary to the results in Figure 2 and Figure 3 showing a greater effect of amyloid at the lower current and short ISI values, Figure 4 shows that the number of statistically significant effects is least in these conditions. This is simply because the variability of the signals in these conditions is much greater than in other conditions as they are closer to the firing threshold of the nerve.

#### 2.1.2. Waveform Data

Figure 5 shows the normalized NAP waveforms averaged over all nerves studied at the same Aβ42 concentration. At the beginning of the experiment, all the waveforms are nearly identical because of the normalization process. However, over time, the amplitude of the NAP declines with only minor changes in the wave shape. The NAP amplitude is highest with no Aβ42 and lowest with Aβ42 concentrations between 70 nM and 700 nM. Figure 6 compares the normalized NAP waveforms at the end of the experiment in the short and long ISI conditions. The NAP in the 1 ms ISI condition is very different from that in the long ISI condition, with lower amplitudes and prolonged peak latencies, although for the most part the effect of Aβ42 is similar. Appendix A displays data in a similar format to that used in Figure 6 and compares the NAP waveforms at the end of the experiment with 4 ms ISIs but 2 mA and 15 mA stimulus currents. The Aβ42 has a much larger effect on the amplitude of the NAP in the 2 mA stimulus condition as compared to the 15 mA condition. Figure 7 and Figure 8 show the relationship between the normalized NAP and the correlation of the voltage at each time point with the Aβ42 concentration. The Spearman R values correlate strongly with the NAP voltage, with the highest correlation at the highest voltages. If the shape of the NAP did not change, but only its amplitude, then the plot of R versus time and the NAP would be similar (Appendix A). This is as would be expected given the previous results that it was amplitude and not any of the shape parameters that varied most significantly with Aβ42 concentration. The lower part of both graphs indicates that the areas where the Spearman rank correlation with the Aβ42 concentration characterized by *p* < 0.01 are near the peak and to a lesser extent the trough (no measurements had FDR < 0.05 according to the BH algorithm). The χ^2^ tests show (Appendix A) that the *p* values < 0.01 in the relationship between the NAP do occur at specific points in the waveform and vary with the interstimulus interval (SEQ from stimulus set 2).

It should be noted that the state of aggregation of the Aβ42 was not assessed, nor was the concentration near the nerve.

## 3. Discussion

These studies demonstrate that Aβ42, when applied to the peripheral nerve in physiologic concentrations, produces changes in the nerve action potential that are readily quantified and characterized. A number of conclusions can be drawn from this data.

### 3.1. Physiologic Effects of Aβ42 on Peripheral Nerve

Although A*β*42 at low concentrations can lower the amplitude of the NAP significantly, there is very little effect on other parameters except when the ISI (inter-stimulus interval) is short or the stimulus intensity is low. If A*β*42 affected the dynamics of sodium activation currents, it would likely change the time required for the NAP to reach peak values even in the long ISI and high stimulus intensity conditions. This could manifest as a change in the rise latency or a change in the velocity or rise amplitude, but none of these are noted. If A*β*42 affected the dynamics of sodium inactivation currents, it would likely change the time required for the NAP to reach trough values, which would manifest as a change in the duration or decline latency or a change in the decline amplitude (Figure 9). Similarly, the absence of any effects on the duration, repolarization latency, or the repolarization amplitude makes it less likely that Aβ42 produces its effects through alterations in the dynamics of potassium channels.

Thus, in this model system, the three most likely effects of low concentrations of Aβ42 would be to: (1) selectively inactivate some sodium channels while others function normally, (2) deactivate some axons without affecting others, or (3) change the passive properties of the axonal membranes. The data from the low stimulus current and short ISI conditions distinguish the first two hypotheses. In the short ISI condition, where there is increased sodium channel inactivation, the changes in NAP amplitude induced by the Aβ42 are greater in the short than in the long ISI condition (Figure 6). In addition, the NAP waveforms shown in Appendix A clearly demonstrate that Aβ42 has a greater effect on the NAP peak-to-peak amplitude at the low stimulus currents than at the high stimulus currents. In these conditions, as in Figure 4, the statistical significance of these effects is less because of the greater variability. Since the effect of the loss of axons would not be ISI- and threshold-dependent, the second explanation is unlikely unless Aβ42 affects axons of different sizes differently.

Is it possible that Aβ42 affects the passive properties of the axonal membrane such as resistance and capacitance? A decrease in membrane capacitance might cause an increase in conduction velocity and a decrease in the stimulation threshold, neither of which are seen. Although an increase of capacitance may increase the stimulation threshold, it would decrease conduction velocity, which is not seen. By the same token, changes in membrane resistance would affect both velocity and threshold. However, in myelinated nerves, if the Aβ42 was more likely to affect the resistance or capacitance of the nodal membranes than the internode, it is possible that the effects on conduction velocity may not be greatly affected. When a brief current I is injected into a (nodal) membrane, the change in voltage associated with this is I/C, where C is the membrane capacitance. Since the membrane must be depolarized to a threshold voltage prior to triggering an action potential, increases in the membrane capacitance would reduce the chance that a given current depolarizes the membrane beyond the threshold. Valincius [42] has shown that the addition of Aβ42 oligomers does increase the capacitance of black lipid membranes. The capacitance of a membrane is given by C = εA/d, where ε is the dielectric constant, A is the area, and d is the membrane thickness. Valincius [42] has suggested that changes in capacitance could be due to changes in the dielectric constant of the membrane produced by pore-related increases in water content, since water has a higher dielectric constant than lipid. They also demonstrated a reduction in the membrane thickness d with the addition of Aβ42. Even though these changes are small in bulk, there may be membrane patches where there is much higher capacitance that could block action potential formation. In support of the role of Aβ42 on membrane properties, Frankel [43] and Sasahara [44] have pointed to the effects that amyloid has on membranes and the modulation of this effect by cholesterol. Furthermore, in hereditary ATTR amyloid peripheral neuropathy, Lai [45] has suggested that amyloid increased internodal capacitance and reduced sodium channel expression. In this same model, Ohashi [46] and Davion [47] have found both axonal and demyelinating features in nerve conduction studies. However, in this study, demyelinating features were not prominent. This may be explained by the fact that the initial effects of amyloid in the ATTR model are on axons; only later does amyloid affect myelin [48], so demyelinating effects might not be seen in this short-term study.

### 3.2. Concentration Effect of Aβ42

At low concentrations, increasing the concentration of Aβ42 reduces the amplitude of the NAP, but at concentrations above 70 nM, increasing the Aβ42 concentration improves the amplitude of the NAP. One possibility is that Aβ42 at higher concentrations may improve neural function. Carillo-Mora [49] has suggested that Aβ42 may have antioxidant, neuroprotective effects and may improve long-term potentiation in the hippocampus. In other studies, Bukanova [50] found a complex “N-shaped” dependence of the Aβ42 effect and suggested that this may be due to an effect on glycine receptors. Kontush has found an antioxidant effect [51] of Aβ42, as has Sinha [52]. Giuffrida [53] found neuroprotective effects of Aβ42 (100 nM) that were dependent on the phosphatidylinositol-3 kinase (PI3-K)/Akt signaling pathway. The latter authors also found that monomeric but not oligomeric Aβ42 could ameliorate N-methyl-D-aspartate (NMDA)-induced neurotoxicity, a phenomenon that was also observed by Niidome [54]. Another possibility suggested by the fact that the NAPs in the presence of high concentrations of Aβ42 are very similar to those recorded in the absence of Aβ42 is that only certain size/conformation aggregates of Aβ42 are toxic and that they form only in a specific concentration range.

### 3.3. Dynamics of Aβ42 Effect

This study also yields information on the dynamics of the effect of Aβ42 on the peripheral nerve. Figure 2 and Figure 3 and Appendix A show that it takes at least 10–15 time periods or 5–8 h for the effect of Aβ42 to fully manifest. Previous studies have shown that aggregation of Aβ42 occurs in a number of different phases [55]: pre-nucleation, post-nucleation, and protofibril elongation/association. All of these processes are slow [56]. Ghosh, in a simulation study [55], suggested that the elongation/association phase occurs on the time scale of an hour or so and that this was highly concentration-dependent, with much more rapid processes at higher concentrations. These simulations were performed with micromolar concentrations of Aβ42, and thus it might be expected that with picomolar concentrations, this aspect of the aggregation process may be very much longer. Walsh [57] and Lomakin [58] found times on the order of tens to hundreds of hours for the formation and elongation of Aβ42 fibrils, even in millimolar concentrations. However, these studies refer specifically to aggregation in solution, and not on the surface of a lipid membrane, which could have very different kinetics, and so these physiologic kinetic studies in this study provide important new information. It is also important to realize that the effect of Aβ42 was not reversed within the 30 min washout period, and so whatever binds the Aβ42 binds it relatively tightly.

Given the slow dynamics and the negative effects of Aβ42 at low concentrations and the positive effects at high concentrations, one might speculate that a possible physiologic response to toxic low concentrations of Aβ42 might be to produce more Aβ42 in order to reduce these toxic effects. If this happens quickly, the effects of the Aβ42 are ameliorated, but if this response is delayed, the toxic effects of Aβ42 become irreversible. This could lead to situations where there is a rapid accumulation of Aβ42 both in patients with and without Aβ42-related injury. This fits with the observation that some individuals have large amounts of amyloid deposition on PET scans without significant negative effects and others have amyloid deposition and very significant negative effects [59]. Some studies have suggested signaling pathways by which this might occur. These include m-RNA-based amplification of amyloid precursor protein production [60], feedback loops involving acetylcholine receptors (α7-nAchR) [61], or complex non-linear dynamics involving the prion protein PRP^C^ and cyclic nucleotides [62].

## 4. Materials and Methods

### 4.1. Experimental Setup

Additional details of the neurophysiologic methods are given in previous papers [28,63,64,65]. Under a protocol approved by the IACUC (Winthrop University Hospital Protocol, WUH-MS#1), a total of 112 nerves from 56 Sprague-Dawley rats (Hilltop, Scottdale, PA, USA) were studied. The rats were male retired breeders with an average age of 32 weeks and an age range of 25–48 weeks. Each sciatic nerve was dissected and placed into a perfusion chamber and stimulated using stainless steel subdermal electrodes arranged in a tripolar array. Experiments were performed with the nerve at 36 °C. The base perfusate was composed of 10 mM HEPES, 110.2 mM NaCl, 17.8 mM NaHCO_3_, 4.0 mM MgSO_4_, 3.9 mM KCl, 3.0 mM KH_2_PO_4_, 1.2 mM CaCl_2_, and 5.5 mM D-glucose, as in previous studies.

### 4.2. Preparation of Aβ42

Aβ42, after the preparation steps noted below, was added to the perfusate 90 min after the start of the experiment. The nine different concentrations of Aβ42 used in this study were: 0 pM, 70 pM, 700 pM, 7 nM, 70 nM, 700 nM, 7 µM, 70 µM, and 700 µM. This brackets the concentrations of Aβ42 measured in the cerebrospinal fluid of 100–200 pM [66] and a concentration felt to be toxic of 1 μM [67], but it also includes concentrations that are much larger [68,69,70], since it is possible that concentrations of Aβ42 may be much higher near the neurons [71,72]—possibly as high as 200 mM [73].

Aβ42 oligomers were created according to the following protocol [74]. Aβ42 (Sigma, St. Louis, MO, USA) was dissolved in 100% hexafluoroisopropanol (HFIP) to a concentration of 1 mM in a glass syringe and then incubated for 15–20 min at 37 °C until the solution was clear. The solution was then partitioned into 10 smaller tubes. For each, the HFIP was evaporated under nitrogen until a peptide film formed at the bottom of the tube. The film was then frozen at −20 °C. Upon use, it was dissolved in 1 N HCL at 4 °C for 24 h. PBS was then added dropwise to a total amyloid concentration of 1 mM. This method was used by Pachara to study amyloid fragments [74] and created rings of amyloid fragments in a β-conformation. Other studies have used related protocols to address the effects of preparation on the structure of the aggregates [75,76].

No imaging was carried out to determine the state of aggregation of Aβ42 in the sciatic nerves or in the perfusion bath. Amyloid preparation was the same in all experiments.

### 4.3. Electrophysiology

Two different sets of stimuli were delivered using a specially developed isolated computer-controlled constant current stimulator. One of the two sets of seven stimuli were delivered every 4 s so that after 8 s, the nerve would have been exposed to both sets of stimuli (Figure 10). The first set consisted of a sequence of stimuli with increasing stimulus intensity, starting at 1 mA and increasing to 2 mA, 3 mA, 4 mA, 6 mA, 10 mA, and 15 mA at 4 ms interstimulus intervals. The second set of stimuli all used a 15 mA stimulus current but had varying interstimulus intervals (ISIs) of: 166 ms, 8 ms, 4 ms, 3 ms, 2 ms, 1.5 ms, and 1 ms, as in Figure 2. Using these two series of stimuli allowed a greater exploration of the effects on the NAP than the more limited protocols used in prior papers. Recordings were made from paired recording electrodes an average of 1 cm away from the stimulating electrodes. Recorded signals were digitized at a 99 kHz/channel, averaged, and stored after 20 averages (approx. 5 s). Each experiment lasted roughly 1000 s. 

### 4.4. Statistics

There are two specific types of analysis performed. One involves the comparison of parameters extracted from the NAP and the other involves the comparison of actual NAP waveforms.

#### 4.4.1. Parametric Analyses

Twelve parameters were abstracted from each NAP, as illustrated in Figure 9. These include the peak-to-peak NAP amplitude, the peak amplitude, the trough amplitude, and the velocity, which is computed as the distance between recording and stimulating electrodes divided by the latency to the NAP peak. There are a number of markers of the NAP shape. The time between the peak and trough is termed “duration”. Another descriptor of the declining phase of the NAP is the time elapsed from the peak to the point where the NAP voltage reaches a value halfway between that of the peak and the trough. This is called the decline latency. The rising phase of the NAP is characterized by the “rise latency”, which is the time between the NAP first reaching an amplitude halfway between the peak and trough and the peak. These indices may be difficult to extract, especially when the NAP trough is small and difficult to mark. Another index of the declining phase (“decline amplitude”) of the NAP is the difference between the amplitude 0.08 ms beyond the peak latency and the peak latency itself. Similarly, the rate of rise of the NAP prior to the peak (“rise amplitude”) can be quantitated as the difference in amplitude of the NAP at the peak and 0.08 ms prior. The recovery period after the trough is characterized by the time it takes the NAP to return to an amplitude halfway between the trough and the baseline as well as the difference in amplitude between the NAP at the trough and 0.16 ms later. The final NAP descriptor is the stimulus response ratio (SRR), which in the first set of stimuli is the peak-to-peak amplitude of each NAP divided by the peak-to-peak amplitude of the last stimulus (largest stimulation current) in the sequence. In the second set of stimuli, the SRR is computed as the ratio between the peak-to-peak amplitude of the given NAP and the peak-to-peak amplitude of the first NAP in the set (longest ISI).

The experiment is divided into 30 min segments labelled sequentially by the variable EXPTTIME. The first two thirty-minute segments are considered time for equilibration, and the mean value of all parameters in the third is used to normalize all of the abstracted parameters so that they are equal to 1 in the third segment. The Aβ42 is added at the beginning of the fourth segment. Each experiment has 36 total 30 min segments, with the last segment starting as the perfusate changes to an Aβ42-free rather than an Aβ42-containing perfusate. Only EXPTTIMES 4–36 are included in the statistical analyses of the abstracted parameters. Although the NAPs are recorded roughly every 4–5 s, only the mean value in each 30 min time period is used in the statistical analysis to reduce the number of variables. The concentration of Aβ42 is represented by the ordinal variable CONC taking on the values 0 (no Aβ42, 40 nerves), 1 (70 pM, 8 nerves), 2 (700 pM, 8 nerves), 3 (7 nM, 8 nerves), 4 (70 nM, 8 nerves), 5 (700 nM, 8 nerves), 6 (7 uM, 8 nerves), 7 (70 uM, 8 nerves), and 8 (700 uM, 16 nerves). The location of a particular stimulus within a sequence of stimuli is called SEQ (taking on values 1–7), and whether the stimulus comes from the sequence of increasing current or shortening ISI is specified by the variable STIM (0—increasing stim current, 1—decreasing ISI). Sequential time points in the NAP waveform are indexed by the variable TIME.

#### 4.4.2. Statistical Testing

In the analysis of the parametric data, there are 112 nerves, 12 parameters per NAP, and 14 NAPs for each of the 33 time periods, so that there are a total of 620,925 measured parameters. In order to simplify the analysis of this massive amount of data, a number of analyses were performed. The first analyses were aimed at determining the relationship between the Aβ42 concentration and the NAP using only the parameters describing the first NAP in the second set at the end of the experiment (EXPTTIME = 36) as a function of the Aβ42 concentration. A simple ANOVA with CONC as the independent factor was performed for each of the 12 parameters with Bonferroni correction for multiple testing so that the significance level of each test was taken at *p* = 0.05/12 = 0.004. Confirmatory testing of significant results was performed using the Kruskal–Wallis non-parametric test at a significance level of 0.1/12 = 0.008. Because of the possibility of a non-linear relationship between the NAP parameters and the Aβ42 concentration, a linear regression analysis was carried out with the concentration and its square as the independent factors.

In order to understand the interactions between CONC, EXPTTIME, and the stimulus amplitude and the interstimulus interval, repeated measures ANOVA was employed. A separate analysis was performed to analyze the effects of interstimulus interval (ISI) using the peak-to-peak NAP amplitude at each of the 7 intervals as the repeated measures variable and EXPTTIME and CONC as the independent factors (Figure 2, Appendix A). Another analysis was used with the amplitude of the NAP at 6 different stimulus intensities as the repeated measures variables and EXPTTIME and CONC as independent factors. Given the non-linear effects of CONC, only concentrations ≤ 0.70 nM were used in these analyses.

#### 4.4.3. Corrections for Multiple Testing

Another approach (Appendix A) which provides more intuitive information uses the Benjamini–Hochberg (BH) procedure (Benjamini and Hochberg, 1995) to identify variables showing a relationship to the Aβ42 concentration associated with a false discovery rate (FDR) of less than 0.05. The *p* value used in this procedure will be probability-associated with the Spearman rank correlation between the parameter values and the Aβ42 concentrations less than or equal to 70 nM. Tables are then formed based on the number of EXPTTIME values for which each parameter in each SEQ within a set are identified as having FDR < 0.05. This not only provides a simple graphical description of the effects of Aβ42 on various aspects of the NAP but also a simple means of testing whether Aβ42 affects one NAP parameter more than another. χ^2^ tests are performed to determine whether the number of significant tests is different for each parameter. In order to make sure that any significant results are not an artifact of the procedure used to identify significant values, the analyses are repeated using all tests with *p* < 0.01.

#### 4.4.4. NAP Waveform Analysis

In addition to studying the parameters abstracted from the NAP, additional insight into the effect of Aβ42 was provided by a direct analysis of the NAP waveform. Since each NAP had a slightly different waveform shape due to the placement of stimulating and recording electrodes, it was important to “normalize” the waveforms prior to the main analysis. This was conducted by taking the averaged NAP from each nerve and stimulus during the first 30 min of the experiment and matching it to the NAP produced by an arbitrarily selected nerve during the same period. Let w_njkl_ (i) be the amplitude of the NAP waveform from the n’th nerve at the i’th time point after the stimulus, j be the value of EXPTIME (in 30 min intervals from the beginning of the experiment), and k represent the k’th stimulus in stimulus set l (1 or 2). The “normalized” waveforms w*_njkl_ (i) are then defined as:w*_njkl_ (i) = a_nkl_w_njkl_ (b_nkl_i + c_nkl_) − d_nkl_(1)
where the parameters ankl (amplitude), bnkl (duration index), cnkl (latency), and dnkl (offset) are chosen to minimize the difference between the normalized waveform and the waveform of the exemplar nerve (n = 1) at the beginning of the experiment (j = 1):
(2)$$I_{nkl}=\sum_{i=1}^{np}v_{i}[w_{11kl}\left(i\right)-w_{11kl}^{*}(i){]}^{2}$$

In this expression, I_nkl_ is the weighted mean square difference between the normalized and template waveforms and np is the number of points in the waveform. The v_i_ are weights chosen to be 1 during the main part of the NAP waveform but take on the value of 0 for the first 10 points to avoid fitting stimulus artifact, and they quickly approach zero after the 100’th data point to reduce the effects of the long tails of the NAP on the fitting procedure. Next, a Spearman rank correlation analysis is performed with the voltage in the normalized waveform at each TIME point as the dependent variable and the Aβ42 concentration as the independent factor with CONC values of 0, 70 pM, 700 pM, 7 nM, and 70 nM (the range of concentrations over which there is a decrease in amplitude with concentration). This generates a correlation value R and probability *p* value for each point in the wave form expressing the degree of correlation between that data point and the Aβ42 concentration. It is important to note that if only certain parts of the NAP waveform change with Aβ42 concentration, only those elements will have significant R values. If only the amplitude of the NAP changes with Aβ42 concentration, then the plot of R vs time will be similar to that of the NAP voltage over time. The degree of correlation between the Spearman R values and the NAP voltage over time is then studied with another Spearman to test the hypothesis that the reduction in NAP amplitude is associated with the Aβ42 effect. A significant correlation indicates that changes in NAP amplitude are associated with Aβ42 concentration in the low concentration range. A second statistical analysis involves determining the number of EXPTIME points in which the time point in waveform SEQ has a *p* < 0.01 in the rank correlation analysis with Aβ42 concentration. χ^2^ testing was used to determine whether the number of significant values is different at different time points in the NAP waveform.

## 5. Conclusions

The in vitro rat sciatic nerve model is sensitive to the effects of physiologic concentrations of Aβ42 and hence may provide a new means of studying the physiologic effects of Aβ42 on axons. The effect of Aβ42 is predominantly on the NAP amplitude and is maximal at concentrations of roughly 70 nM, with either lower or higher concentrations having less effect on the NAP amplitude. In this model system, the effects of Aβ42 became maximal 5–8 h after exposure and did not reverse during a 30 min washout period.

## Figures and Tables

**Figure 1 ijms-24-14488-f001:**
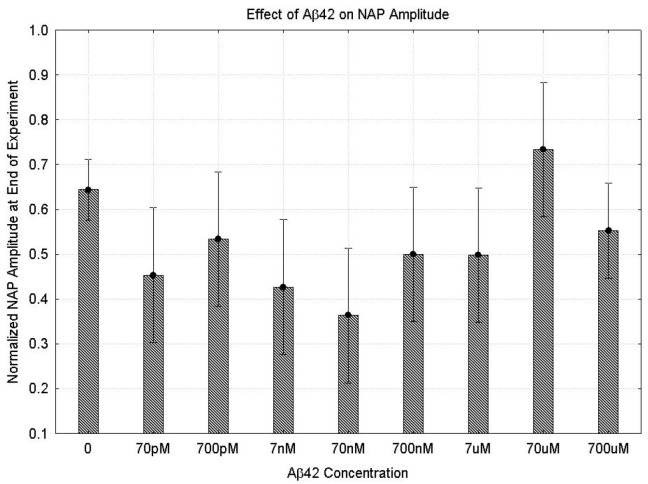
Changes in the peak-to-peak NAP amplitude at the end of the experiment as a function of Aβ42 concentration. All data are taken from the first stimulus in the second set of stimuli (166 ms ISI and 15 mA stimulus current). The amplitude is normalized so that the amplitude of the NAP at the beginning of the experiment is one. The bars indicate the interquartile range. The differences between different conditions are significantly different (ANOVA (F (8103) = 3.08, *p* = 0.004)).

**Figure 2 ijms-24-14488-f002:**
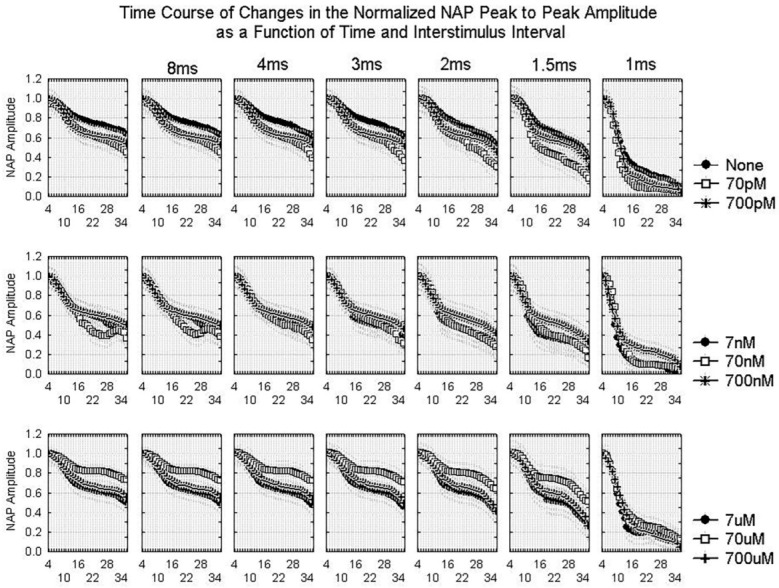
Averaged time course of changes in the normalized NAP peak-to-peak amplitude during the course of an experiment for different interstimulus intervals and different Aβ42 concentrations with a 15 mA stimulus current. Because of the normalization procedure, all NAP peak-to-peak amplitudes are set at one for the beginning of the experiment. The different curves represent either the control condition (“none”) or responses in the presence of various concentrations of Aβ42. The x-axis represents the number of 30 min periods from the beginning of the experiment. Repeated measures ANOVA shows significant effects of EXPTTIME, CONC, and ISI as well as significant CONC*ISI, ISI*EXPTTIME, and CONC*EXPTTIME interactions, all with *p* < 0.001 (Appendix A).

**Figure 3 ijms-24-14488-f003:**
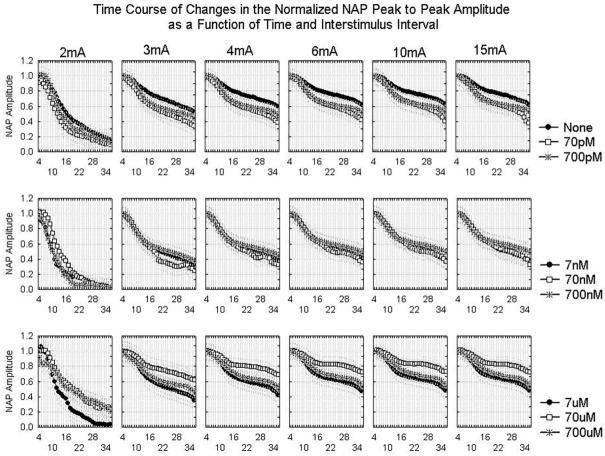
Averaged time course of changes in the normalized NAP peak-to-peak amplitude during an experiment for different stimulus currents and different Aβ42 concentrations using a 4 ms interstimulus interval. As in Figure 2, the x-axis is the number of 30 min periods from the beginning of the experiment. Each curve represents data from either the control (“none”) or conditions with varying Aβ42 concentrations. Repeated measures ANOVA shows significant effects of EXPTTIME, CONC, and AMP as well as significant CONC*AMP, AMP*EXPTTIME, and CONC*EXPTTIME interactions, all with *p* < 0.001 (Appendix A).

**Figure 4 ijms-24-14488-f004:**
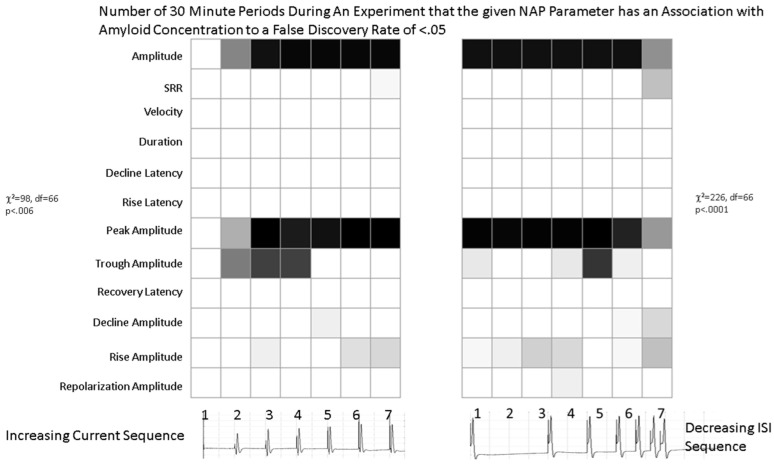
Graphical tables indicating the number of 30 min time periods during which the Spearman rank correlation analysis shows a significant (FDR < 0.05) between the NAP characteristic on the left of the table and the Aβ42 concentration and for the specific stimulus shown on the bottom of the table. A white color in the table element indicates that the NAP characteristic was not affected by the concentration of Aβ42. Darker colors indicate conditions where there are more significant relationships between the NAP characteristic and the Aβ42 concentration. The χ^2^ test tests the question as to whether the pattern of significant effects is truly random. The different rows are different NAP parameters and the different columns represent different stimuli within a set.

**Figure 5 ijms-24-14488-f005:**
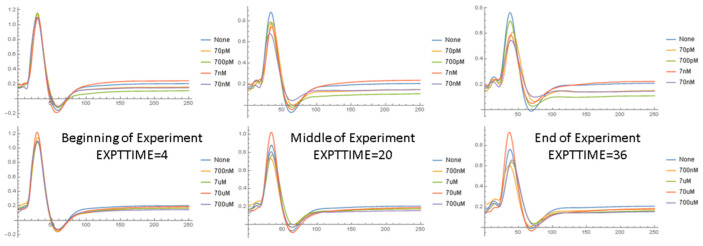
Normalized NAP waveforms averaged over all nerves in same Aβ42 concentration category at the beginning of the experiment, middle of the experiment, and end of the experiment for the long (166 ms) inter-stimulus interval at different concentrations of Aβ42. Each curve represents a different concentration of Aβ42 and “none” is the control case in which no Aβ42 is added.

**Figure 6 ijms-24-14488-f006:**
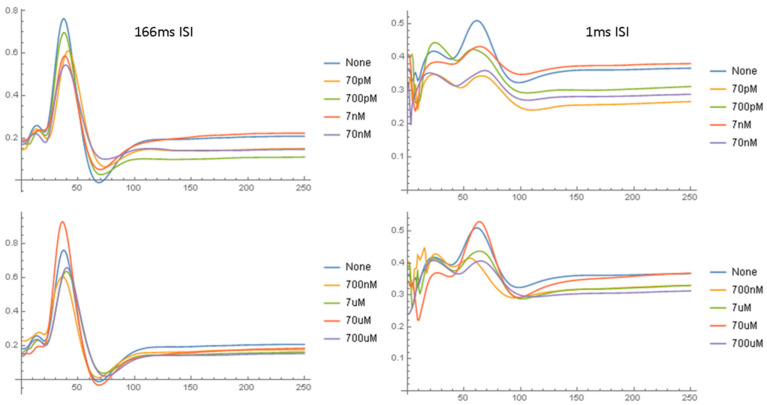
Normalized NAP waveforms averaged over all nerves in same Aβ42 concentration category at the end of the experiment (EXPTTIME = 36) for the long (166 ms) and short (1 ms) ISI conditions at 15 mA stimulus current. The NAPs from the short ISI (interstimulus interval) conditions are distorted due to the presence of residual sodium channel inactivation at those intervals.

**Figure 7 ijms-24-14488-f007:**
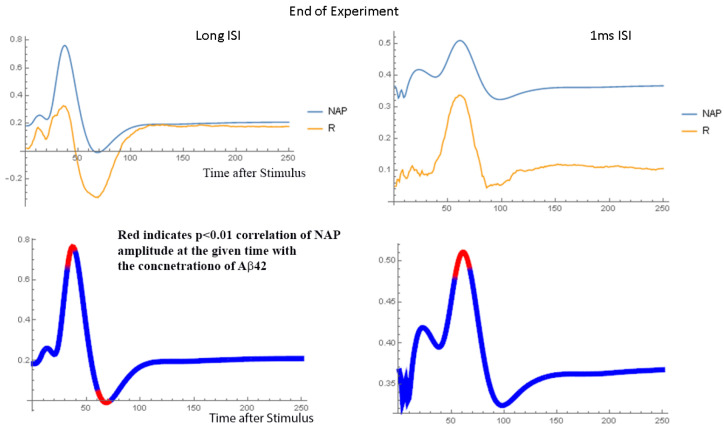
Shows the normalized NAP at the beginning of the experiment along with the R value associated with the Spearman rank correlation analysis when applied to each time point in the NAP waveform. The lower set of figures shows on the NAP waveform the times when the R value associated with the Spearman rank correlation analysis is significant with *p* < 0.01. Short and long interstimulus interval at 15 mA current. In the lower plots, the red areas highlight the parts of the NAP waveform that are most highly correlated with the concentration of Aβ42. The observation that the plot of R versus time is similar to that of the NAP indicates that the presence of Aβ42 does not significantly change the shape of the NAP.

**Figure 8 ijms-24-14488-f008:**
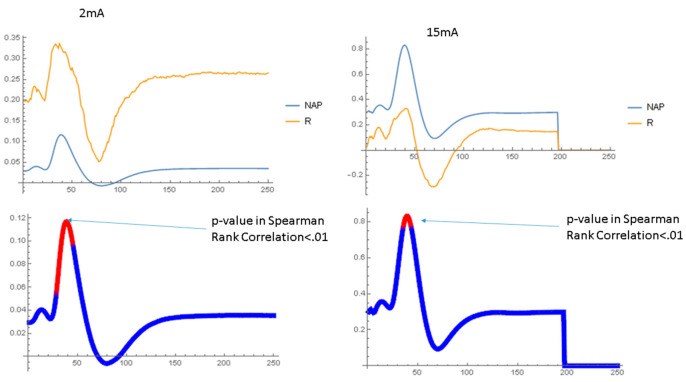
Shows the normalized NAP at the beginning of the experiment along with the R value associated with the Spearman rank correlation analysis when applied to each time point in the NAP waveform. The lower set of figures shows on the NAP waveform the latencies at which the R value associated with the Spearman rank correlation analysis between the amplitude at that time and Aβ42 concentration is significant with *p* < 0.01. Low and high stimulation currents at 4 ms interstimulus interval.

**Figure 9 ijms-24-14488-f009:**
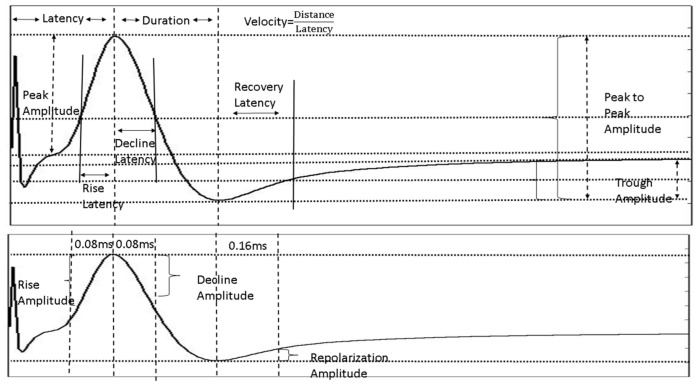
Parameters extracted from each NAP tracing by the automated marking algorithms. These are the NAP characteristics used in Figure 4.

**Figure 10 ijms-24-14488-f010:**
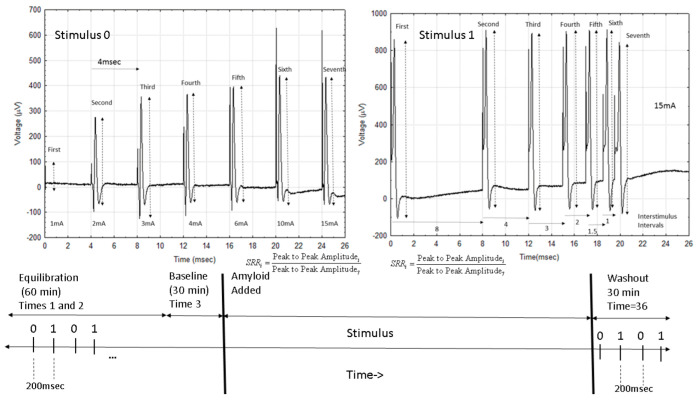
Outline of the experiment. The top part of the figure shows the two different stimulus sets used with the first, keeping the interstimulus interval at 4 ms and increasing the stimulus current. The second maintains a constant stimulus current but gradually shortens the interstimulus interval. The lower graph shows an outline of each experiment and the fact that stimulus set 0 is delivered first followed by stimulus set 1 every 200 ms during the experiment. It also shows the initial equilibration period and the time that the amyloid is added into the fluid, bathing the nerve.

## Data Availability

The full dataset is available from the corresponding author, upon motivated request.

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
