# Peer review of "Amyloid-β Effects on Peripheral Nerve: A New Model System"

_ijms, 2023, doi:10.3390/ijms241914488_

Round 1
Reviewer 1 Report
Reviewer Comments
The present research article has explored the possibility that the in-vitro sciatic nerve model can be used to study the effects of Aβ42 on the peripheral nerve. This model system has the advantage that it has been studied extensively and the nerve action potential (NAP) provides a sensitive, easy-to-obtain, and easily quantifiable marker of the function of the nerve.
The research paper is well-organized and discusses the results very well. However, a few suggestions are here to incorporate.
Scientific comments
1. Line 62-64, further additions.....without Ab42 is not understandable. Rewrite the sentence.
2. Line 64-66, results are vaguely described in this section.
3. Figure 1, graphical representation can be improved.
4. Line 225-226, discuss briefly to show connectivity with previous lines.
5. Line 260-261, discuss these lines briefly. Which signaling pathways are involved?
Minor corrections/suggestions
6. References are not in IJMS format. Please check the format and revise accordingly.
7. Line 14, Low is repeated 2 times. delete one.
8. Check citation patterns [1-3] and correct them throughout the MS.
9. Lines 28-30, Sentence is not correct. Rewrite the sentence.
10. Check line 73. Is the sentence complete?
11. Line 121, What does it mean by same format?
12. Line 240, Ghosh or Ghosh et al? Check for other references too.
13. Line 146, remove hyphen from word associ-ated.
Author Response
9/18/23
Editor
MDPI, International Journal of Molecular Sciences
Section: Molecular Neurobiology
Special Issue: Novel Hypotheses for Dementia and Neurodegenerative Diseases: From Molecular Mechanisms to Therapies
Re: Amyloid-β Effects on Peripheral Nerve: A New Model System
Editors:
Please find enclosed our detailed responses to the comments of the reviewers and a revision of the above captioned manuscript by Mark M. Stecker, Ankita Srivastava and Allison B. Reiss.
Reviewer 1
The research paper is well-organized and discusses the results very well. However, a few suggestions are here to incorporate.
The authors thank the reviewer for their comments.
Line 62-64, further additions.....without Ab42 is not understandable. Rewrite the sentence.
Changes made as suggested.
Line 64-66, results are vaguely described in this section.
Changes made as suggested.
Figure 1, graphical representation can be improved.
The figure legend was updated and the left title was updated to indicate that the amplitudes are normalized and instead of amyloid Ab42 is noted. The color of the bars was changed to allow visualization of the interquartile range more clearly.
Line 225-226, discuss briefly to show connectivity with previous lines.
Changes made as suggested
Line 260-261, discuss these lines briefly. Which signaling pathways are involved?
Changes made as suggested
Minor corrections/suggestions
References are not in IJMS format. Please check the format and revise accordingly.
We double checked the styles recommended in the information for authors and our reference format seems to fit with those as it does with papers from IJMS on line. Please let us know about specific changes. The IJMS format in refworks does not reproduce the format selected only the MDPI-cells format seems to fit.
Line 14, Low is repeated 2 times. delete one.
Change made as suggested.
Check citation patterns [1-3] and correct them throughout the MS.
As above we double checked the information for authors and some published papers and our reference style seems to be the same. We would be grateful for more specific information.
Lines 28-30, Sentence is not correct. Rewrite the sentence.
Re-written as suggested
Check line 73. Is the sentence complete?
Changes made as suggested.
Line 121, What does it mean by same format?
Changes made to clarify the sentence as suggested
Line 240, Ghosh or Ghosh et al? Check for other references too.
We reviewed the information for authors and there was no suggestion as to which is preferred. Looking at a few published papers from the journal we didn’t find any use of the et. al. except in the references. We would be glad to hear which is preferred.
- Line 146, remove hyphen from word associ-ated.
We checked after the revisions and did not find this. It may have been an auto generated hyphen inserted because of the justification
We think that the extensive revisions we have made to the paper on the suggestion of the referees has improved the manuscript significantly.
Mark Stecker MD, PhD
Reviewer 2 Report
The authors used Sciatic nerve to follow the dose-dependent (?) changes of Aβ in physiological phenomena with high sensitivity. The authors are analyzing various parameters with the aim of constructing an experimental system to follow the changes in physiological phenomena in a highly sensitive and dose-dependent (?) manner using the Sciatic nerve. However, many of the methods are explained in a way that is quite difficult to understand the purpose of using the methods, and although we know that statistics is done, the explanation is too inadequate for what the statistics can say. The figure legends are also lacking in specificity, and there are many parts that are incomprehensible, which makes the paper feel immature. Of course, it is possible that this reviewer is not the right person for this paper, but should not he/she aim more at communicating research to others in an easy-to-understand manner?
I also fear that what is claimed in Abst may not be accomplished in the experiments conducted in this paper, and I will explain this below.
l Conclusion is just a summary of the results of the experiment. In the end, the authors should clearly state what they have found out and what they think it can be used for.
l It is analytically detailed, but my opinion is that there is not much information in the paper because the Aβ conditions used are all over the place and whether they are monomers or oligomers has not been considered.
l There are not many control experiments, and it will be necessary to verify to what extent this phenomenon is specific to Aβ42, for example, whether such a phenomenon does not occur with Aβ40.
l As for the claim that peripheral nerves are more sensitive, I am skeptical because there is no comparison in this paper and it is not clear whether there is a dose dependence.
l I am not familiar with mathematical models and may have misunderstood, but the section on statistics is too long and it is difficult to understand what is being explained. I would have liked to have seen the section on statistics be more readable, e.g., with titles for each topic.
l And what is ultimately important in these models is whether there is a statistically significant difference? And is the number of n sufficient for statistical treatment? This should be clearly stated in the legend of each figure, but in some figures this explanation is missing.
l Figure 1 is particularly striking, but first of all, the legend ends in the middle. In the text, it is mentioned that multiple comparisons are made, but it is not clear what is being compared and what is being discussed in terms of increase or decrease. Furthermore, if there is a difference, shouldn't the level of significance for the object of comparison be stated in detail? Also, as a highly sensitive observation system, it is necessary to have a certain degree of dose dependence, but I believe that such statistics have not been compiled. Another serious flaw is that the number of trials in the experiment is not clearly stated.
l It is very possible that I do not understand the statistics in Figures 2-3, but it should be more clearly shown whether there is a difference or not at each Aβ concentration compared to the control. Can you be more specific about what you are trying to explain by the statistics?
l Figure 4 is also completely unintelligible. For example, what do the numbers 1-7 mean, and what do the different shades of color in the squares indicate? The authors should strive to provide information that is more generally needed, rather than targeting a specialized audience.
l I know from reading the text that Figures 5 and 6 are fiddling with statistical calculations, but in the end it is not clear what is being compared to what and where the significant differences are, I don't think the n numbers are specified, but it is not a description of statistics that I know much about, so an explanation of the statistics used should be added.
l There is a notation in the method chapter stating that Oligomer was used, but there may be no data.
The discussion part was, well, theoretical, but I thought that the hypothesis we ended up getting would not work as a paper unless we went deeper into the hypothesis.
I think a little more should be said about the validity of this paper as a model, such as how close it is to actual Alzheimer's disease and how the results of this paper can contribute to it. Is it really highly sensitive?
Author Response
9/18/23
Editor
MDPI, International Journal of Molecular Sciences
Section: Molecular Neurobiology
Special Issue: Novel Hypotheses for Dementia and Neurodegenerative Diseases: From Molecular Mechanisms to Therapies
Re: Amyloid-β Effects on Peripheral Nerve: A New Model System
Editors:
Please find enclosed our detailed responses to the comments of the reviewers and a revision of the above captioned manuscript by Mark M. Stecker, Ankita Srivastava and Allison B. Reiss.
Reviewer 2
The authors used Sciatic nerve to follow the dose-dependent (?) changes of Aβ in physiological phenomena with high sensitivity.
The authors do not claim “high sensitivity” just “sensitivity”.
The authors are analyzing various parameters with the aim of constructing an experimental system to follow the changes in physiological phenomena in a highly sensitive and dose-dependent (?) manner using the Sciatic nerve.
To be specific, we are using the well studied in-vitro rat sciatic nerve model but do not make any claims about “high sensitivity”.
However, many of the methods are explained in a way that is quite difficult to understand the purpose of using the methods, and although we know that statistics is done, the explanation is too inadequate for what the statistics can say.
We have used many citations throughout the paper to the extensive work that has been done with this model. Much of what is done here is in keeping with and is an extension of previous work. More specific comments will allow focused improvements.
The figure legends are also lacking in specificity, and there are many parts that are incomprehensible, which makes the paper feel immature. Of course, it is possible that this reviewer is not the right person for this paper, but should not he/she aim more at communicating research to others in an easy-to-understand manner?
Specific comments will lead to more focused changes in the manuscript. We have gone through and extensively improved the figure legends and the figures as suggested to improve clarity.
I also fear that what is claimed in Abst may not be accomplished in the experiments conducted in this paper, and I will explain this below.
The abstract speaks of the effect of Ab42 on the NAP and the time course of those changes. These are clearly demonstrated in the figures.
l Conclusion is just a summary of the results of the experiment. In the end, the authors should clearly state what they have found out and what they think it can be used for.
What we found out in the research was exactly the facts listed in the conclusion section. As in the first line of the conclusion, we feel the research can be used to study the physiologic effects of Ab42. In response to the comments of the reviewer we have emphasized this in the first sentence of the conclusion.
l It is analytically detailed, but my opinion is that there is not much information in the paper because the Aβ conditions used are all over the place and whether they are monomers or oligomers has not been considered.
We do not understand what the reviewer means by stating that the Ab conditions used are “all over the place”. We applied the Ab42 prepared as documented in different concentrations to the in-vitro sciatic nerve model. It is true that we are not sure the exact conformation of the Ab in solution or on the surface of the nerve. To say that this was not considered is not accurate. The discussion speaks in detail about the possibilities. As the first research study to use this approach, it was our intent to first determine whether there would be effects in this model system. Future investigations with more detailed imaging or other size measurement procedures would be needed to answer that important question. In order to make this more clear we did add a sentence to the methods section to clarify this.
l There are not many control experiments, and it will be necessary to verify to what extent this phenomenon is specific to Aβ42, for example, whether such a phenomenon does not occur with Aβ40.
The reviewers comments here are less than clear. There have been extensive controls done in two different venues. First, this model has been used many times in the literature to explore many aspects of nerve physiology and the references are included. Second, the condition “none” or a concentration of 0 of the Ab42 is the control in this experiment. The reviewer’s point about the potential different actions of Ab40 would be important and may be explored in future experiments.
l As for the claim that peripheral nerves are more sensitive, I am skeptical because there is no comparison in this paper and it is not clear whether there is a dose dependence.
Again, the reviewer’s comments are less than clear. The authors never made a claim that the in-vitro sciatic nerve model was more sensitive to the effects of Ab42 than any other model. It is just novel and different. The purpose of the paper was just to show that Ab42 had an effect in this model and do some basic characterization. The reviewer states that it is “not clear whether there is a dose dependence”—Figure 1 clearly shows the effects of different concentrations of Ab42 on the NAP and this effect is statistically significant. The pictures of the NAP’s under different circumstances confirm this as well. It would be useful if the reviewer might state any particular reason why they do not accept the data.
l I am not familiar with mathematical models and may have misunderstood, but the section on statistics is too long and it is difficult to understand what is being explained. I would have liked to have seen the section on statistics be more readable, e.g., with titles for each topic.
We have modified the statistics section based on the comments of the reviewer to include multiple subsections and hope that it improves readability.
l And what is ultimately important in these models is whether there is a statistically significant difference? And is the number of n sufficient for statistical treatment? This should be clearly stated in the legend of each figure, but in some figures this explanation is missing.
Much of the information on statistics requested by the reviewer is already present in the results section. However, we did add this information to the legend for figure 1. We also provided additional information for the number of nerves used in each case in the results section.
l Figure 1 is particularly striking, but first of all, the legend ends in the middle. In the text, it is mentioned that multiple comparisons are made, but it is not clear what is being compared and what is being discussed in terms of increase or decrease. Furthermore, if there is a difference, shouldn't the level of significance for the object of comparison be stated in detail? Also, as a highly sensitive observation system, it is necessary to have a certain degree of dose dependence, but I believe that such statistics have not been compiled. Another serious flaw is that the number of trials in the experiment is not clearly stated.
The level of significance was clearly stated in the first paragraph of the results section using the ANOVA and Kruskal-Wallis tests. The graph clearly shows that it is the NAP amplitude at the end of the experiment that is being compared. We have made some modifications in the figure and the figure legend to improve readability. The number of studies compared is implicit in the degrees of freedom for the F test used for the ANOVA but we added annotation about the number of nerves tested for each Ab42 concentration. Supplementary table 2 shows the results of linear regression analysis of NAP amplitude on CONC and CONC*CONC showing significant effects only for the amplitudes. Thus, we have used 3 different statistical tests to confirm the significance of the effect of CONC on NAP amplitude.
l It is very possible that I do not understand the statistics in Figures 2-3, but it should be more clearly shown whether there is a difference or not at each Aβ concentration compared to the control. Can you be more specific about what you are trying to explain by the statistics?
Figure 1 shows the concentration effects on the amplitude and Supplementary figure 2 shows a similar graph for some of the other NAP characteristics. As for Figures 2 and 3, we felt that the graphs themselves were sufficient to show the significant effects discussed in the paper. Repeated measures ANOVA is always complicated and limited but based on the comments of the reviewer we performed a repeated measures ANOVA on the data in Figures 2 and 3 (Contained in supplementary tables 3 and 4) demonstrating significant interactions between interstimulus interval and Ab42 concentration and stimulus intensity and Ab42 concentration that support the patterns seen visually.
l Figure 4 is also completely unintelligible. For example, what do the numbers 1-7 mean, and what do the different shades of color in the squares indicate? The authors should strive to provide information that is more generally needed, rather than targeting a specialized audience.
We have improved the figure legend. As in that figure the numbers 1-7 indicate the stimulus from which the data is taken and that is shown below the table on the left the sequence is of increasing stimulus amplitude and on the right of decreasing interstimulus interval. It is very simple, darker blocks indicate a stronger interaction between Ab42 concentration and the NAP parameter measured for the given stimulus. This is another way of demonstrating the the main effect of amyloid is on the NAP amplitude. Supplementary tables 1 and 2 show this in different ways. We feel that showing the result using different approaches is important. Supplementary figure 1 also provides a graphic illustration about the construction of the tables shown in figure 4.
l I know from reading the text that Figures 5 and 6 are fiddling with statistical calculations, but in the end it is not clear what is being compared to what and where the significant differences are, I don't think the n numbers are specified, but it is not a description of statistics that I know much about, so an explanation of the statistics used should be added.
Figures 5 and 6 are not simply “fiddling”. As pointed out in the revised methodology section one can analyze a waveform by extracting characteristics such as duration, amplitude, velocity, etc. It is also possible to analyze the effect of Ab42 on the NAP at every point after the stimulus. Again, this shows that the effect of Ab42 is not to change the shape of the NAP as much as to change its amplitude. We provided an analysis of this issue in more detail under some special conditions in the addended supplementary data.
l There is a notation in the method chapter stating that Oligomer was used, but there may be no data.
An annotation was made clarifying this as suggested by the reviewer.
The discussion part was, well, theoretical, but I thought that the hypothesis we ended up getting would not work as a paper unless we went deeper into the hypothesis.
Although it might have some theoretical components, they are clearly identified and the discussion helps to place the data obtained in this study in context with other studies.
I think a little more should be said about the validity of this paper as a model, such as how close it is to actual Alzheimer's disease and how the results of this paper can contribute to it. Is it really highly sensitive?
The peripheral nerve is not a model of actual Alzheimer’s disease but it can be used to study some of the physiologic effects of amyloid that might be involved. We have added additional text in the introduction discussing the relationship between nerve injury and Alzheimer’s disease. The authors never claim that the method is “highly” sensitive only that it is sensitive.
We think that the extensive revisions we have made to the paper on the suggestion of the referees has improved the manuscript significantly.
Mark Stecker MD, PhD
Round 2
Reviewer 2 Report
I understand much better now that you have added explanations, but even with this revision there are still some points that remain unclear.
There is a notation for oligomer formation in the Materials and Methods, but there is no word oligomer in the Results, so it is unclear where it was used.
Where exactly is the figure?
Please clearly identify and explain the experiments in which you used oligomer in the Results.
Also, it would be better to clearly show how monomers and oligomers actually react differently in this analysis system.
Since Aβ42 does not exist physiologically on its own, I think it is important to point out the possibility of structural changes to show its usefulness as a model.
Another thing that is troubling is that the decrease in NAP may occur at pathological rather than physiological concentrations.
I think physiological concentrations should be defined in the citation.
I also mentioned that physiological Aβ40 tends to be more abundant and less cohesive, so I thought it was important to distinguish between monomers and oligomers to see if the same phenomenon occurs with Aβ40 as a control.
In interpreting 3.3, is it correct to say that under the conditions of this study, aggregates can form at all concentrations, and the author believes that this is one of the possibilities that causes the decrease in NAP?
I think it is important to note that it takes time to suppress NAP, but it is difficult to see this idea until the discussion, so I thought it would be good to mention it in the abstract or introduction.
However, I felt that other hypotheses could be proposed.
Aβ42 may trigger some kind of signal in a delayed manner, etc.
I thought the description of neuroprotection at 3.2 Aβ42100nM and the benign effect on NAP was a leap.
I am not sure that it is safe to say that a decrease in NAP is neurotoxic, and it would be better to just explain it at the level of NAP regulation.
I mistakenly thought you wanted to claim high sensitivity because Abst said 'strongly' concentration dependent.
So what makes this "strong"?
The other point is that it seems to be a contradiction in terms to say that there is a strong concentration dependence when it is non-linear.
Usually when we think of concentration dependence we think of linearity, so it would be less misleading to change the wording.
Again, my overall impression is that the study of structural changes in Aβ is weak.
This could be improved by clearly defining oligomer and performing experiments.
However, as I interpret the author's results, it seems that there are variations in oligomer, so I think that what kind of oligomer was used should be defined and used in the experiment.
Author Response
9/20/23
Editor
MDPI, International Journal of Molecular Sciences
Section: Molecular Neurobiology
Special Issue: Novel Hypotheses for Dementia and Neurodegenerative Diseases: From Molecular Mechanisms to Therapies
Re: Amyloid-β Effects on Peripheral Nerve: A New Model System
Editors:
Please find enclosed our detailed responses to the second set of comments from Reviewer 2 and a revision of the above captioned manuscript by Mark M. Stecker, Ankita Srivastava and Allison B. Reiss.
The focus of this paper is whether Ab42 would affect peripheral nerve and what effects it might have as a starting point for future investigations. Most of the reviewer’s comments are asking for new experiments that we agree would be interesting for future research but our paper being the first to address the issue of peripheral nerve physiology and Ab42 we wanted first to show that there was an effect before embarking on more complex investigations. In any good research there are always new issues that can be addressed but in this case we have a number of interesting results in a novel system and so the additional experiments suggested by the reviewer are not necessary before publication. Just as an example few if any of the studies on in-vivo imaging of amyloid in humans or the studies of the the use of Ab42 as a biomarker in humans determine the state of aggregation of the Ab42. This is an interesting and important point but does not detract from the value of those studies.
Reviewer 2 comments
I understand much better now that you have added explanations, but even with this revision there are still some points that remain unclear.
We are thankful that the extensive revisions made have improved the manuscript.
There is a notation for oligomer formation in the Materials and Methods, but there is no word oligomer in the Results, so it is unclear where it was used.
In the previous revision of the document the term oligomer is used 3 times not just once as noted by the reviewer. In addition, the term aggregation was used 5 times and the term aggregate was used once. In addition there was the sentence added to the previous version “No imaging was carried out to determine the state of aggregation of Ab42 in the sciatic nerves or in the perfusion bath.” That although it did not use the word oligomer referred to the state of aggregation .
However, in order to respond to the reviewer an additional sentence was added to the results as follows: “It should be noted that the state of aggregation of the Ab42 was not assessed nor was the concentration near the nerve.” We did not use the term oligomer here as we felt that mentioning the state of aggregation was sufficient.
Where exactly is the figure?
It is difficult to respond to this comment unless the reviewer specifies which figure is being referred to.
Please clearly identify and explain the experiments in which you used oligomer in the Results.
In the methods section we detailed how we prepared the Ab42. We have now added additional comments to the comments in the previous version that we did not assess the state of aggregation. However, we did add information about the previous works from which we adopted the preparation techniques that were used and their observations on the type of aggregates formed after this preparation.
Also, it would be better to clearly show how monomers and oligomers actually react differently in this analysis system.
This is a paper focusing on the physiology of peripheral nerve. We agree that the suggestions provided by the author would be interesting future studies but the fact that we do not study aggregation does not make the work uninteresting. As a case in point, the significant and important studies done on in-vivo amyloid imaging or on Ab42 as a biomarker do not address the state of aggregation specifically but are still important.
Since Aβ42 does not exist physiologically on its own, I think it is important to point out the possibility of structural changes to show its usefulness as a model.
This is now pointed out in numerous locations within the paper.
Another thing that is troubling is that the decrease in NAP may occur at pathological rather than physiological concentrations. I think physiological concentrations should be defined in the citation.
The previous version had citations to this effect but other citations were added and the concentrations spelled out in more detail as suggested. As per the Raskatov article cited in the paper there may be over 12 orders of magnitude of potentially relevant amyloid concentrations.
I also mentioned that physiological Aβ40 tends to be more abundant and less cohesive, so I thought it was important to distinguish between monomers and oligomers to see if the same phenomenon occurs with Aβ40 as a control.
Again the primary target of this study is on peripheral nerve physiology. It is not a physical chemical study of aggregation in amyloid fragments. These are certainly interesting and important issues that can be dealt with in future studies.
In interpreting 3.3, is it correct to say that under the conditions of this study, aggregates can form at all concentrations, and the author believes that this is one of the possibilities that causes the decrease in NAP?
Again, this is not a study of aggregation kinetics. Our study was about physiology. What the reviewer suggested is possible.
I think it is important to note that it takes time to suppress NAP, but it is difficult to see this idea until the discussion, so I thought it would be good to mention it in the abstract or introduction.
This is clearly stated in the results section around line 84 and in supplementary figure 3.
However, I felt that other hypotheses could be proposed.Aβ42 may trigger some kind of signal in a delayed manner, etc.
For any set of data there are a large number of possible hypothesis, we focused on just a few in this initial observation. As you can see in the discussion around lines 250 we mention some of the factors that are in the reviewer’s comment.
I thought the description of neuroprotection at 3.2 Aβ42100nM and the benign effect on NAP was a leap.
We cited a reference which discussed this issue and so is not a leap that we are making without literature support.
I am not sure that it is safe to say that a decrease in NAP is neurotoxic, and it would be better to just explain it at the level of NAP regulation.
We never stated that a decrease in the NAP is neurotoxic. Besides the references that discuss neurotoxicity of Ab the only time the neurotoxicity is used in the paper is to refer to studies done to understand the neurotoxicity of oxaliplatin [25] and dust from the world trade center [26].
I mistakenly thought you wanted to claim high sensitivity because Abst said 'strongly' concentration dependent. So what makes this "strong"?
Based on the comment of the reviewer we removed the unclear term “strong” and replaced it with the term “significantly”.
The other point is that it seems to be a contradiction in terms to say that there is a strong concentration dependence when it is non-linear. Usually when we think of concentration dependence we think of linearity, so it would be less misleading to change the wording.
Of course, the reviewer is well aware that the total effect that any intervention may have on a measurable quantity may be very large whether the relationship is linear or non-linear. As in the example shown below in both cases whether the line is red or blue there is a large effect of concentration. In addition I would not take the narrow view that concentration dependence means linearity. This is a common problem where a drug may cause beneficial effects at one concentration and toxicity at another.
Again, my overall impression is that the study of structural changes in Aβ is weak.
We reiterate that the purpose of the study was NOT to study structural changes in Ab but to determine if peripheral nerve could be used as a model system in which to study the effects of Ab. This is the foundational question. Once this has been decided future studies can deal with more complex issues
This could be improved by clearly defining oligomer and performing experiments.
Those would be interesting subsequent experiments.
However, as I interpret the author’s results, it seems that there are variations in oligomer, so I think that what kind of oligomer was used should be defined and used in the experiment.
As in the paper and noted above, we used a standardized methodology adopted by others so that if future experiments were done on the state of aggregation of the Ab, this would be a starting point.
Mark M. Stecker MD, PhD, FASM, FACNS, FAAN
Round 3
Reviewer 2 Report
I will respect the authors' comments.
Please confirm just this point. Are authors saying that the Aβ used in all the experiments in this study was processed in such a way that oligomers were formed?
My understanding was that the monomers (that means not treating to form oligomer) were processed, so apologies if I misunderstood.
However, given the current method and the way the text is written, I find that point rather difficult to understand.
4.2 dictated creation method of oligomer. so authors were using oligomers in this study, right?
If there is a distinction between monomer and oligomer usage, I thought it needed to be mentioned.
If all experiments have been treated with oligomer formation, that should also be noted.
Author Response
9/21/23
Editor
MDPI, International Journal of Molecular Sciences
Section: Molecular Neurobiology
Special Issue: Novel Hypotheses for Dementia and Neurodegenerative Diseases: From Molecular Mechanisms to Therapies
Re: Amyloid-β Effects on Peripheral Nerve: A New Model System
Editors:
Please find enclosed our detailed responses to the second set of comments from Reviewer 2 and a revision of the above captioned manuscript by Mark M. Stecker, Ankita Srivastava and Allison B. Reiss.
Reviewer 2 comments
Please confirm just this point. Are authors saying that the Aβ used in all the experiments in this study was processed in such a way that oligomers were formed?
The references on which the protocol was based [74] produced rings of amyloid fragments in the b-conformation. So, yes oligomers as stated on line 268 of the methods section.
My understanding was that the monomers (that means not treating to form oligomer) were processed, so apologies if I misunderstood.
However, given the current method and the way the text is written, I find that point rather difficult to understand.4.2 dictated creation method of oligomer. so authors were using oligomers in this study, right?
As above, the references on which the protocol was based [74] produced rings of amyloid fragments in the b-conformation.
If there is a distinction between monomer and oligomer usage, I thought it needed to be mentioned.
All experiments were carried out with amyloid prepared as in section 4.2
If all experiments have been treated with oligomer formation, that should also be noted.
A notation was made that in all experiments the preparation of the amyloid was the same
Mark M. Stecker MD, PhD, FASM, FACNS, FAAN